# Reductive Catalytic Fractionation of Spruce Wood over Ru/C Bifunctional Catalyst in the Medium of Ethanol and Molecular Hydrogen

Oxana P. Taran [1,2,*], Angelina V. Miroshnikova [1,2,*], Sergey V. Baryshnikov [1], Aleksandr S. Kazachenko [1,2], Andrey M. Skripnikov [1,2], Valentin V. Sychev [1,2], Yuriy N. Malyar [1,2] and Boris N. Kuznetsov [1,2]

1   Institute of Chemistry and Chemical Technology SB RAS, Krasnoyarsk Science Center, Siberian Branch, Russian Academy of Sciences, Akademgorodok 50, Bld. 24, 660036 Krasnoyarsk, Russia
2   Department of Non-Ferrous Metals and Materials Science, Siberian Federal University, pr. Svobodny 79, 660041 Krasnoyarsk, Russia
*   Correspondence: taran.op@icct.krasn.ru (O.P.T.); miroshnikova.av@icct.krasn.ru (A.V.M.)

**Abstract:** Reductive catalytic fractionation (RCF) has emerged as an effective lignin-first biorefinery strategy to depolymerize lignin into tractable fragments in high yields. Herein, we propose the RCF of spruce wood over a Ru/C bifunctional catalyst in the medium of ethanol and molecular hydrogen to produce monomeric phenolic compounds from lignin, polyols from hemicelluloses, and microcrystalline cellulose. This contribution attempts to elucidate the role of the Ru/C bifunctional catalysts characteristics. The results clarify the particular effect of the carbon support acidity, catalyst grain size, content and dispersion of Ru on the effectiveness of lignin and hemicelluloses extraction and the yields of liquid and gaseous products. The most efficient catalysts for RCF of spruce wood, providing high yields of the monomeric phenols, glycols, and solid product with content of cellulose up to 90 wt%, bear 3 wt% of Ru with a dispersion of 0.94 based on an acidic oxidized graphite-like carbon support Sibunit®, and having a grain size of 56–94 μm. The Ru/C catalysts intensify the reactions of hydrodeoxygenation of liquid products from lignin. The main phenolic monomers are 4-propyl guaiacol, 4-propenyl guaiacol, and 4-propanol guaiacol. We explored the effect of the process temperature and time on the yield and composition of the liquid, solid, and gaseous products of spruce wood RCF. The optimal trade-off between the yields of phenolic monomers (30.0 wt%), polyols (18.6 wt%) and the solid product containing 84.4 wt% of cellulose is reached at 225 °C and 3 h over the most acidic Ru/C catalyst.

**Keywords:** spruce wood; fractionation; reduction; Ru/C catalysts; ethanol; hydrogen; liquid products; phenols; polyols; cellulose

## 1. Introduction

The depletion of fossil resources and $CO_2$ emissions negatively affecting the environment require new methods for the use of renewable plant feedstock, including agricultural waste, to ensure environmental safety and social and economic sustainability. In recent years, there has been increased interest in deep processing of renewable plant biomass due to the fact that the use of biomass, in contrast to fossil organic raw materials, does not change the $CO_2$ balance in the atmosphere [1–3]. Wood is a widespread and available resource for the production of various chemical products and biofuels. The main components of wood (cellulose, hemicelluloses, and lignin) form a complex biocomposite complicating chemical conversion of wood into high-demand products.

The most large-scale industrial process in the chemical conversion of wood is the production of cellulose [4]. In conventional pulp production processes, hemicelluloses and lignin are extracted as by-products; besides, technical lignins are highly condensed and contain sulfur that hinders their further processing. Hence, the development of new

processes to ensure processing of all main components of lignocellulosic biomass into various chemical products and liquid biofuels is an important topic [5,6]. Among such processes is organosolv delignification of wood using organic solvents [7,8]. A lower degree of condensation of organo-soluble lignins than technical lignins and the absence of sulfur facilitates their catalytic processing into required phenolic and aromatic substances [5,9]. However, when organosolv lignins are isolated, the undesirable polymerization processes can occur to decrease their yield and reactivity. These circumstances make it urgent to develop methods of lignin depolymerization which eliminate the additional stage of its isolation from lignocellulosic material.

The most promising approaches to the complex conversion of wood are based on the methods of oxidative [10–12] and reductive [12–14] catalytic fractionation of wood biomass. Oxidative catalytic fractionation leads to the formation of valuable products: microcrystalline cellulose, aromatic and aliphatic acids, and vanillin. In particular, the reductive catalytic fractionation (RCF) of lignocellulosic biomass ensures depolymerization of lignin to form liquid hydrocarbons while preserving the main part of cellulose [4,6]. Hemicelluloses depolymerize partially to form soluble products. In this case, the products formed during the partial depolymerization of hemicelluloses can be separated from the lignin products by extraction with dichloromethane and water [15]. Processes of reductive depolymerization of lignin are achieved over metallic catalysts (Pd/C, Ni-Raney, Ni/C, and Ni/Al$_2$O$_3$), over the reductive hydrogen donor agent [16,17]. The hydrogen donor agents like aliphatic alcohols (methanol, ethanol, or isopropanol) [11–13] and formic acids [14,15] are used. Supported noble metals (Pt, Pd, and Ru) catalysts show highest activity in these processes, thus providing high yields of phenolic monomers and high-quality cellulose [13,18–20]. The nature of catalysts and solvents used, as well as the process conditions (temperature, time, etc.) determine the yield and composition of phenolic monomers from lignin [15].

The use of Ru/C, Pt/C, and Rh/C catalysts leads to propyl-substituted methoxyphenols S/G (propyl guaiacol/syringol) as the main monomer compounds in the liquid products of softwood and hardwood hydrogenation [21–23]. The Pd/C or Ni/C catalysts give propanol-substituted monomers (4-propanol guaiacol and syringol) [23,24]. Over the Ru/C catalyst, the main products of the RCF process are 4-propanol guaiacol in the butanol/water medium [18] and 4-propyl guaiacol in the methanol medium [23].

The resulting methoxyphenols can be used in various fields; in particular, 4-propyl guaiacol can serve as a component of epoxy resins and polycarbonates [25,26] and 4-propanolguaiacol and its derivatives with the antioxidant and anti-inflammatory activity have prospects for application in medicine [27]. Polyols, e.g., propylene glycol and ethylene glycol, are used to produce foams, elastomers, and adhesives, as well as pharmaceuticals, antifreezes, and solvents [28,29]. The cellulose product can be used to obtain high-demand chemicals (ethanol, esters, levulinic and lactic acids, 5-hydroxymethylfurfural and others) [30–32]. Microcrystalline cellulose finds application in the pharmaceutical, food, cosmetic, and perfume industries and in production of sorbents [33]. In addition, it can be a basis of nanocellulose materials and composites [34].

The use of bifunctional solid acidic catalysts with platinum metals results in intensification the lignin depolymerization and enhances the yield of monomeric phenol derivatives [13,35]. A bifunctional catalyst containing 3% of Ru on an acid-modified graphite-like carbon Sibunit® has been previously shown to exhibit the high activity in the hydrogenolysis of wood and ethanol lignin of birch [12,32,36] and abies [37]. However, the content and nature of acidic species on the surface of the carbon support can affect both the acidic properties of the catalyst and the deposition of metal particles on its surface. This issue has not been studied earlier.

Herein, we explored the influence of characteristics of the Ru/C bifunctional catalysts (the acidity of a carbon support, catalyst gain size, content and dispersion of Ru) as well as the conditions (temperature and time) of the process of spruce wood RCF on the yield, composition, and structure of the liquid, solid, and gaseous products. The conditions

for the optimal balance between the yields of monomeric phenols and polyols as well as cellulose product have been found. One of the main forest forming tree species in Siberia and in the north of the European part of Russia—the spruce *Picea obováta* [38]—was chosen as the object of this study.

## 2. Results

### 2.1. Characterization of Ru/C Catalysts

We chose a commercial synthetic graphite-like carbon material of the Sibunit® family as the catalyst support. The large surface area, high mechanical strength and the mesoporous structures make Sibunit materials the promising liquid-phase catalysts. Original Sibunit grains (1.0–1.6 mm in size) and powder (56–94 μm in particle size) were used as supports. In order to create surface oxygenate species, Sibunit was oxidized with gas mixture saturated with water vapor ($O_2/N_2 = 1/5$) at 400, 450 and 500 °C.

Table 1 shows $N_2$ adsorption, TEM, and $pH_{pzc}$ data on the oxidized carbon supports and Ru/C catalysts. After the oxidative treatment of almost all the carbon supports used, except the sample oxidized at 450 °C, the Brunauer–Emmett–Teller specific surface areas ($S_{BET}$) decreased. The deposition of ruthenium onto the carbon support surface also leads to a decrease in the specific surface area and pore volume (Table 1) due to blocking of some pores of the supports by particles of the active component [9,39].

**Table 1.** Characteristics of the carbon supports and ruthenium catalysts.

| No. | Support/Catalysts | Label | $S_{BET}$, m$^2$/g | $V_{pore}$, cm$^3$/g | $\langle d \rangle_{pore}$, nm | $pH_{pzc}$ [a] | Ru Particle Size, nm [b] | | | | $D_{Ru}$ |
| | | | | | | | $d_{min}$ | $d_{max}$ | $<d_1>$ | $<d_s>$ | |
|---|---|---|---|---|---|---|---|---|---|---|---|
| 1 | Sib-4 [c] | S | 375 | 0.55 | 5.87 | 7.59 | - | - | - | - | - |
| 2 | 3%Ru/Sib-4 [c] | 3RS | 321 | 0.43 | 5.45 | 8.01 | 0.60 | 2.73 | 1.22 ± 0.01 | 1.48 | 0.88 |
| 3 | Sib-4-granular [d] | Sg | 364 | 0.51 | 5.66 | 7.66 | - | - | - | - | - |
| 4 | 3%Ru/Sib-4-granular | 3RSg | 273 | 0.32 | 4.77 | 8.05 | 0.76 | 3.46 | 1.42 ± 0.02 | 1.71 | 0.77 |
| 5 | Sib-4-ox-400 [c] | S400 | 332 | 0.42 | 5.06 | 6.88 | - | - | - | - | - |
| 6 | 3%Ru/Sib-4-ox-400 [c] | 3RS400 | 300 | 0.37 | 5.01 | 7.12 | 0.66 | 3.00 | 1.19 ± 0.01 | 1.40 | 0.94 |
| 7 | Sib-4-ox-450 [c] | S450 | 380 | 0.53 | 5.66 | 5.33 | - | - | - | - | - |
| 8 | 1%Ru/Sib-4-ox-450 [c] | 1RS450 | 368 | 0.52 | 4.80 | 6.06 | 0.52 | 1.79 | 1.06 ± 0.03 | 1.27 | 1.03 |
| 9 | 3%Ru/Sib-4-ox-450 [c] | 3RS450 | 341 | 0.50 | 5.88 | 6.89 | 0.52 | 2.37 | 1.13 ± 0.01 | 1.39 | 0.94 |
| 10 | Sib-4-ox-500-granular [d] | S500g | 287 | 0.37 | 5.14 | 3.34 | - | - | - | - | - |
| 11 | 3%Sib-4-ox-500-granular [d] | 3RS500g | 233 | 0.28 | 4.80 | 6.44 | 0.69 | 3.14 | 1.30 ± 0.01 | 1.53 | 0.85 |

[a] $pH_{pzc}$ is pH at the zero charge point (PZC); [b] $d_{min}$ and $d_{max}$ are the minimum and maximum particle size, $<d_1> = \Sigma d_i/N$ is the average particle size, $<d_S> = \Sigma d_i^3/\Sigma d_i^2$ is the weight average particle size, and $D_{Ru}$ is the Ru dispersion; [c] the grains are 56–94 μm in size; [d] the grains are 1.0–1.6 mm in size.

The interaction of negatively charged surface functional groups of the oxidized support with ruthenium ions during the catalyst preparation prevents sintering of ruthenium particles in the course of the high temperature reduction of the catalyst [40]. Thus, the oxidative treatment of the support results in an increase in the content of sorption-active groups on the surface, which helps to reduce the size of metal particles (Figure 1).

The temperature elevation during the oxidative treatment of the carbon support leads to an increase in the support and the catalyst acidities (Table 1). The deposition of ruthenium onto the surface of the initial and oxidized supports raises $pH_{pzc}$. Note that a slightly alkaline reaction (pH) is characteristic of the catalyst slurry based on the nonoxidized Sibunit-4 support.

Thus, the bifunctional ruthenium catalysts bearing Ru nanoparticles can be prepared via deposition of $Ru(NO)(NO_3)_3$ on the surface of the oxidized Sibunit-4 carbon support. The prepared series of the Ru/C catalysts with different grain size, Ru content (1 and 3 wt%), dispersion (0.77–1.00), and acidity (pH 3.34–8.05) were tested for hydrogenation of spruce wood in an ethanol medium.

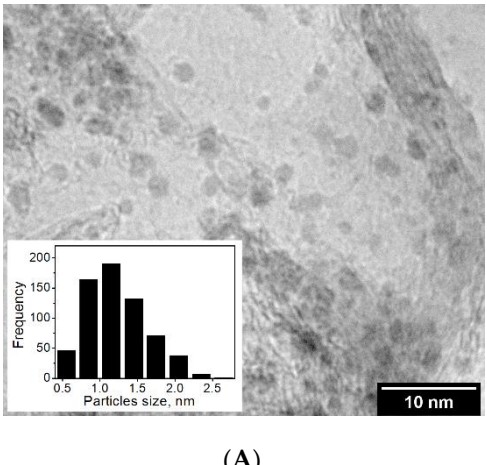

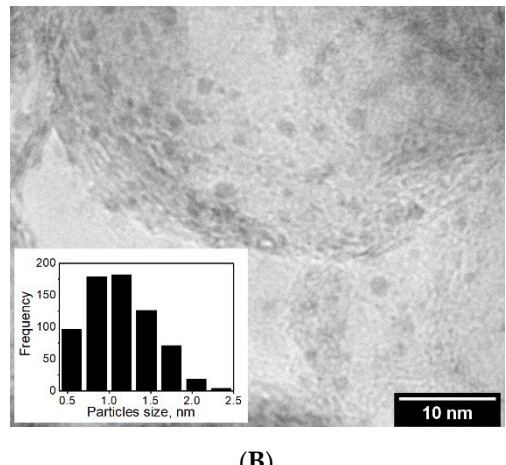

(**A**)    (**B**)

**Figure 1.** High-resolution transmission electron microscopy images and Ru-particle side distribution for the Ru (3%)/Sib-4 (**A**) and Ru (3%)/Sib-4-ox-450 (**B**) catalysts.

### 2.2. Yield and Composition of the Spruce Wood RCF Products over the Ru/C Catalysts

Non-catalytic hydrogenation of spruce wood in ethanol at 225 °C and an initial $H_2$ pressure of 4 MPa for 3 h yields liquid products (23.8 wt%), gaseous products (4.4 wt%), and solid products (67.7 wt%) (Table 2). The aqueous phase contains up to 3.5 wt% of polyhydric alcohols including ethylene glycol, propylene glycol, 1,2-butanediol, and pentane-1,2,5-triol as products of decomposition of the carbohydrate constituent of the wood. The solid product contains 59.4 wt% of cellulose, 24.4 wt% of lignin, and 16.2 wt% of hemicelluloses (Table 2).

The yield of the liquid products of wood RCF over the 3RS450 catalyst increases to 36 wt% and the contents of phenolic monomers in the organic phase and polyols in the aqueous phase increase to 26 and 18.6 wt%, respectively. The cellulose content in the solid product increases to 84.4 wt% at a simultaneous decrease in the contents of hemicelluloses and lignin to 7.0 and 8.6 wt%, respectively (Table 2).

**Table 2.** The results of RCF of spruce wood at 225 °C for 3 h over different Ru catalysts.

| Catalyst | Product Yield, wt% | | | | | Solid Product Composition, wt% | | |
|---|---|---|---|---|---|---|---|---|
| | Solid | Liquid | Gaseous | Phenolic Monomers [c] | Polyols [d] | Hemicelluloses | Lignin | Cellulose |
| no | 67.7 | 23.8 | 4.4 | 1.5 | 3.5 | 16.2 | 24.4 | 59.4 |
| 3RS450 | 49.5 | 36.0 | 10.6 | 26.0 | 18.6 | 7.0 | 8.6 | 84.4 |
| 3RS400 | 54.3 | 32.5 | 9.0 | 30.0 | 13.9 | 14.8 | 7.6 | 77.6 |
| 3RS500g [b] | 54.7 | 27.5 | 12.2 | 12.0 | 11.3 | 15.8 | 9.3 | 74.9 |
| 3RS [a] | 57.4 | 30.0 | 9.0 | 22.0 | 12.5 | 14.1 | 9.8 | 76.1 |
| 3RSg [a,b] | 57.6 | 29.0 | 7.8 | 15.7 | 10.6 | 16.5 | 12.1 | 71.4 |

[a] Non-oxidized support; [b] a grain size of 1.0–1.6 mm; [c] the yield from the lignin weight in wood; [d] the yield from the polysaccharide weight in wood.

The gain size affects significantly the yield and composition of the liquid, solid, and gaseous products of spruce wood hydrogenation. The granular catalysts (1.0–1.6 mm fraction) are less efficient to the process of RCF of spruce wood than the powder catalysts (56–94 μm fraction). When used, the monomer and polyol yields are lower by a factor of 1.5–2 over the granular catalysts than over the powder catalysts, obviously, due to diffusion restrictions inherent in liquid-phase catalytic processes.

Figure 2 shows yields of the main RCF products obtained without catalysts and over different catalysts. The remaining monomer products were found in trace amounts. The maximum content (30 wt%) of monomer phenols was observed in the liquid products over the 3RS400 catalyst and the maximum content (18.6 wt%) of polyols over the 3RS450 catalyst. Notably, 4-propenyl guaiacol dominates among the phenolic products of the non-

catalytic hydrogenation of spruce wood (Figure 2A). The main component of the phenol products in the RCF is 4-propyl guaiacol at a yield of 18.1 wt% over the 3RS450 catalyst and 23.9 wt% over the 3RS400 catalyst (Figure 2A). In the non-catalytic hydrogenation of wood, the yield of glycols is no more than 3.5 wt% (Figure 2B). In the liquid products of the RCF, the yield of glycols, mainly ethylene glycol and propylene glycol, rises sharply (by a factor of 2.9) (Figure 2B).

We observed a trend to an increase in the total yields of monomer methoxyphenols and polyols with the increasing dispersion of ruthenium in the catalysts. The higher Ru dispersion accounts for better active center accessibility to promote to the higher overall catalytic activity [41] (Figure 3).

Elemental analysis of the liquid products of spruce wood hydrogenation reveals intensification of the deoxygenation reactions over the ruthenium catalysts (Figure 4).

The liquid products of the spruce wood RCF at 225 °C contain a lower amount of oxygen and a higher amount of hydrogen and carbon compared to those in the initial wood; this leads one to conclude about more intensive deoxygenation reactions over the ruthenium catalysts. The minimum oxygen content and the maximum carbon content are observed in the liquid products of wood RCF over the 3RS400 and 3RS450 catalysts. According to the Van Krevelen diagram, the atomic ratio O/C of liquid products of non-catalytic and catalytic hydrogenation of spruce wood is much lower than in the original wood. The most significant decrease in O/C and the highest H/C ratio in liquid products were observed over the 3RS450 catalyst.

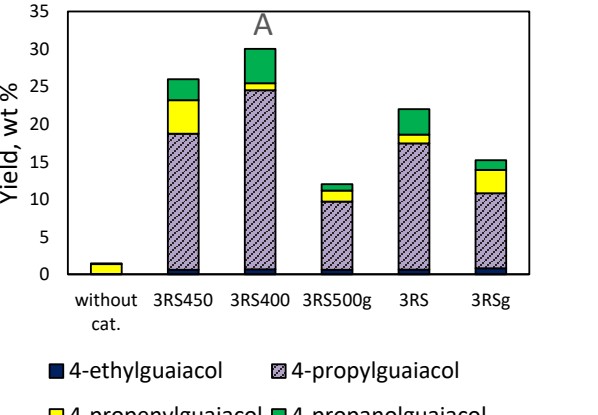 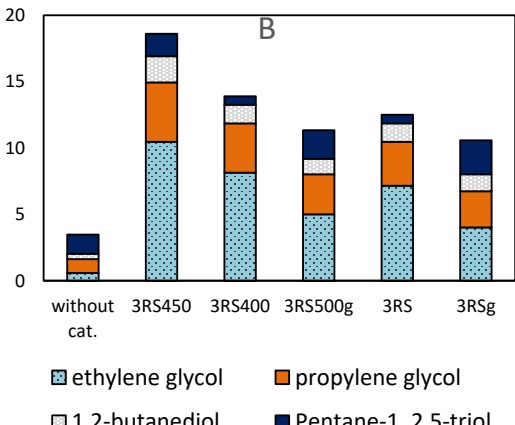

**Figure 2.** The yields of monomers from lignin (**A**) and polyols from polysaccharides (**B**) over the Ru catalysts.

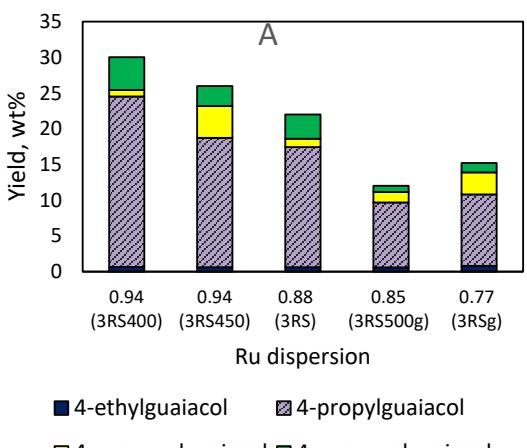 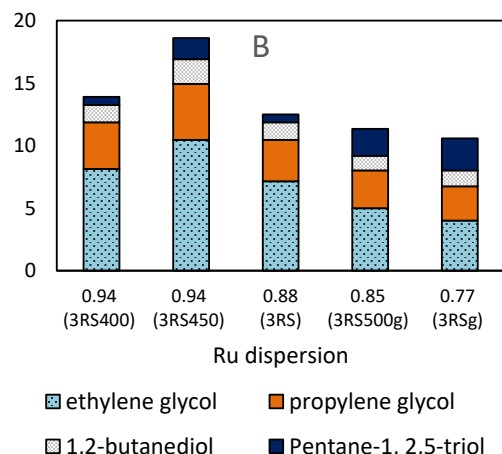

**Figure 3.** The influence of Ru dispersion in the catalyst on the yields of monomers from lignin and (**A**) polyols from polysaccharides (**B**).

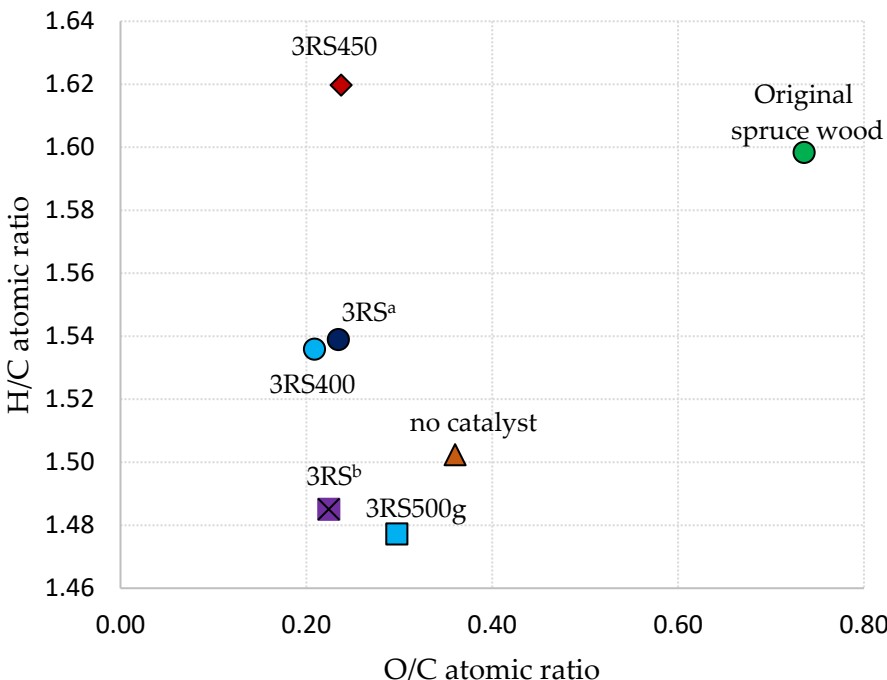

**Figure 4.** Van Krevelen diagram for the liquid products of spruce wood RCF at 225 °C for 3 h over different catalyst ([a] Original (non-oxidized) support; [b] the grain size is 1.0–1.6 mm).

Carbon oxides (4.4 wt%) and trace methane (less than 0.1 wt%) were detected in gaseous products of non-catalytic hydrogenation. The methane content increases significantly (up to 4.5 wt%) in the gaseous products over the ruthenium catalysts. Ruthenium is known to catalyze $CO_2$ conversion to methane. The usual reaction temperature range is from 200 to 450 °C depending on the catalyst, support and the reaction conditions. So, it is reasonable to assume such pathways in our case. However, the main pathway of methane formation is catalytic hydrocracking of aliphatic structural fragments of lignin [42].

We examined the effect of the ruthenium content in the catalyst on the yield and composition of the products of spruce wood RCF at 225 °C (Table 3).

**Table 3.** The effect of ruthenium content in the catalyst on the yield and composition of products of spruce wood RCF at 225 °C for 3 h.

| Catalyst | Product Yield, wt% | | | | | Solid Residue Composition, wt% | | |
|---|---|---|---|---|---|---|---|---|
| | Solid | Liquid | Gaseous | Phenolic Monomers [a] | Polyols [b] | Hemicelluloses | Lignin | Cellulose |
| no | 67.7 | 23.8 | 4.4 | 1.5 | 3.5 | 16.2 | 24.4 | 59.4 |
| 1RS450 | 60.0 | 28.7 | 9.1 | 23.1 | 14.2 | 10.2 | 9.3 | 80.5 |
| 3RS450 | 49.5 | 36.0 | 10.6 | 26.0 | 18.6 | 7.0 | 8.6 | 84.4 |

[a] The yield from the lignin weight in wood; [b] the yield from the polysaccharide weight in wood.

Compared to the non-catalytic process, a 15-fold increase in the content of monomer phenolic compounds (Figure 5A) and a 4-fold increase in the content of polyols is observed in the spruce wood RCF over the catalyst containing 1 wt% of Ru. At the RCF of spruce wood, the contents of ethylene glycol and propylene glycol increase and the content of pentane-1,2,5-triol decreases (Figure 5B). At a higher Ru content in the catalyst (3 wt%), the yield of the liquid products of wood RCF increases up to 36.0 wt% and the content of phenolic monomers decreases to 26.0 wt% and the content of polyols to 18.6 wt%. While the yield of the solid product drops down, the cellulose content increases to 84.4 wt% (Table 3).

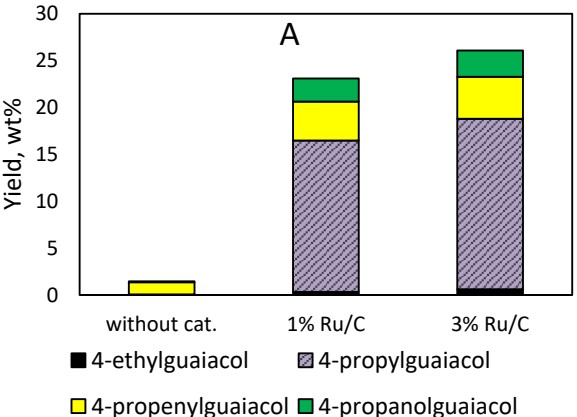
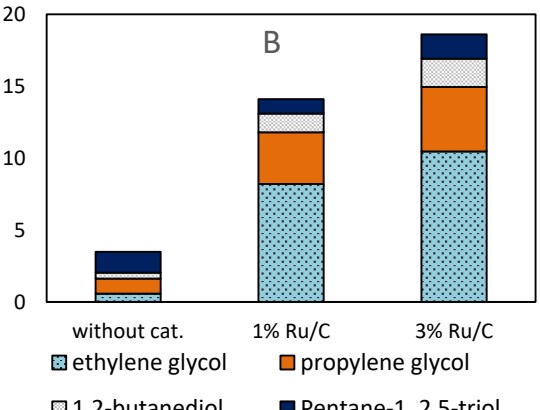

**Figure 5.** The effect of Ru content in the catalyst on the yield of phenolic monomers (**A**) and polyols (**B**) after spruce wood RCF at 225 °C for 3 h.

The influence of the metal content in the Ru/C catalyst on the molecular weight distribution (MWD) of the liquid products of spruce wood RCF at 225 °C for 3 h was studied using GPC (Figure 6).

Among the liquid products of the non-catalytic hydrogenation of spruce wood, there are fractions of organosolv lignin ($M_W$ = 1000–10,000 g mol$^{-1}$) and products, mainly dimers, of its partial depolymerization. With the catalysts, the profile of the MWD curve of the liquid products changes significantly to indicate a change in their composition. The MWD curves of the liquid products of RCF contain an intense peak at $M_W$ of ~150 g/mol corresponding to the monomer compounds. An increase in the amount of ruthenium in the Ru/C catalyst to 3 wt% leads to a decrease in the content of oligomers with 2–4 units ($M_W$ ~ 400–600 g/mol) and an increase in the content of the monomer compounds in the liquid products (Figure 6, Table 4).

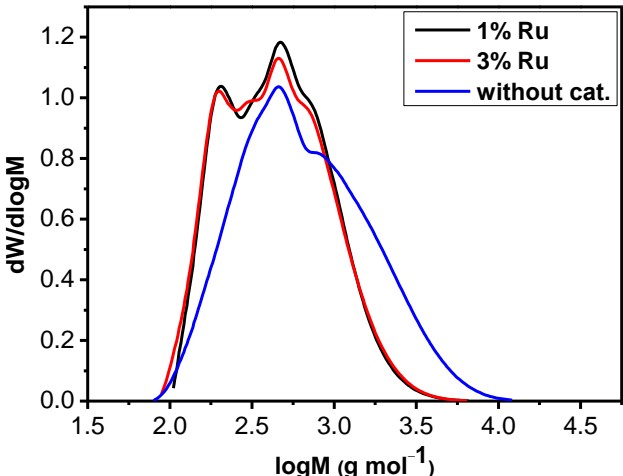

**Figure 6.** The effect of Ru content on the molecular weight distribution of the liquid products of spruce wood RCF at 225 °C for 3 h.

**Table 4.** Effect of the Ru content on molecular weight characteristics of the liquid products of spruce wood RCF at 225 °C for 3 h.

| Catalyst | $M_n$ [a], g mol$^{-1}$ | $M_w$ [b], g mol$^{-1}$ | PD [c] |
|----------|----------|----------|--------|
| no | 458 | 1001 | 2.186 |
| 3RS450 | 337 | 578 | 1.715 |
| 1RS450 | 351 | 580 | 1.652 |

[a] The number average molecular weight; [b] the weight average molecular weight; [c] the polydispersity $M_w/M_n$.

An increase in the ruthenium content in the catalyst from 1 to 3 wt% leads to a slight increase in the yield of gaseous carbon oxides with a noticeable rise in the yield of methane (Figure 7).

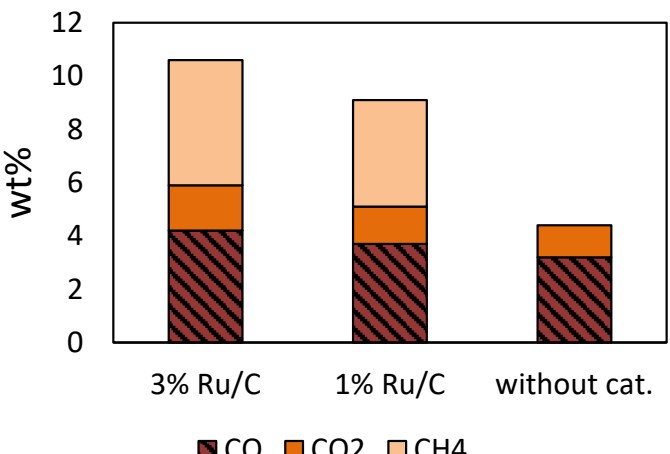

**Figure 7.** The effect of Ru content on the yield and composition of the gaseous products of spruce wood RCF at 225 °C for 3 h.

*2.3. Influence of Temperature on the Yield and Composition of the Products of Spruce Wood RCF over the 3RS450 Catalyst*

To study the effect of temperature on the yield and composition of the spruce wood RCF products, we chose a 3RS450 catalyst that provides the highest yield of the liquid products (Table 5).

**Table 5.** The effect of temperature on the yield and composition of the products of spruce wood RCF (3 h) over the 3RS450 catalyst.

| Temperature, °C | Product Yield, wt% | | | | | Solid Residue Composition, wt% | | |
|---|---|---|---|---|---|---|---|---|
| | Solid | Liquid | Gaseous | Phenolic Monomers [a] | Polyols [b] | Hemicelluloses | Lignin | Cellulose |
| 200 | 69.0 | 19.6 | 6.7 | 15.6 | 10.2 | 18.6 | 14.5 | 56.7 |
| 225 | 49.5 | 36.0 | 10.6 | 26.0 | 18.6 | 7.0 | 8.6 | 84.4 |
| 250 | 41.0 | 32.0 | 18.1 | 35.0 | 19.1 | 4.4 | 5.2 | 90.4 |

[a] The yield from the lignin weight in wood; [b] the yield from the polysaccharides weight in wood.

The liquid product yield increases from 19.6 to 36 wt% over this catalyst as the process temperature rises from 200 to 225 °C. The temperature rises up to 250 °C leads to a decrease in the yield of the liquid products down to 32 wt% at a simultaneous increase in the contents of phenolic monomers and polyols. The solid product yield decreases from 69 to 41 wt% with the hydrogenation temperature rise from 200 to 250 °C (Table 5).

The process temperature elevation from 200 to 250 °C leads to an increase in the cellulose content in the solid product of wood RCF from 56.7 to 90.4 wt%, while the contents of lignin and hemicelluloses decrease from 13.2 and 17.6 to 5.2 and 4.4 wt%, respectively (Table 5). The content of phenolic monomers in the liquid products also increases from 15.6 to 35 wt% (Figure 8A). We observed an increase in the content of 4-ethylguaiacol (1), 4-propylguaiacol (2), and 4-propenylguaiacol (3) and reduction of the content of 4-propanolguaiacol (4) with the hydrogenation temperature rising from 200 to 225 °C. 4-Propanolguaiacol appears to convert into 4-propylguaiacol upon elimination of the γ-OH group with the rise of the RCF temperature [43].

The most noticeable increase (from 10.2 to 18.6 wt%) of the polyol yields is observed when the temperature rises from 200 to 225 °C; the further temperature elevation to 250 °C does not change significantly this yield (Figure 8B).

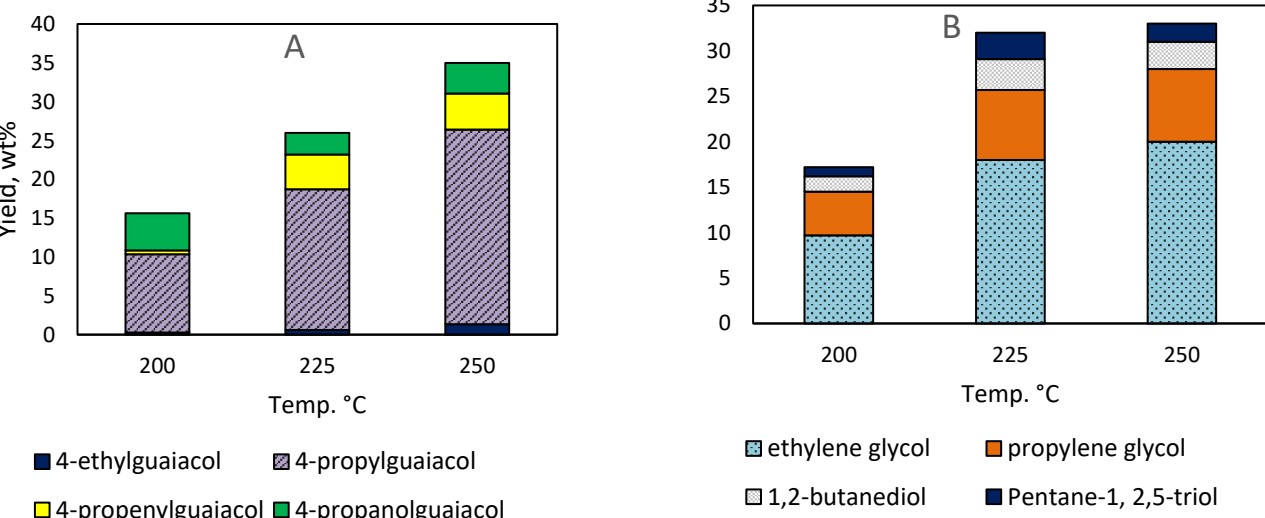

**Figure 8.** The effect of temperature of spruce wood RCF over the 3RS450 catalyst on the yield of phenolic monomers (**A**) and polyols (**B**).

From GPC data, the rise of the wood RCF temperature from 200 to 250 °C facilitates the reduction of the average molecular weight $M_w$ of the liquid products from 578 to 502 g/mol (Table 6). The intensity of the MWD curves of the liquid products at the region corresponding to the monomeric compounds with MW ≈ 150 g/mol increases with the rise of the RCF temperature. In addition, a curve shoulder assigned to oligomeric fragments of lignin depolymerization narrows noticeably. However, at 250 °C, the intensity increases at the region of dimers with MW = 350–400 g/mol to indicate the contribution of the repolymerization reactions. The polydispersity PD of the liquid products passes through a maximum at 225 °C to increase from 1.614 to 1.715 and then decreases to 1.679 at 250 °C (Table 6, Figure 9).

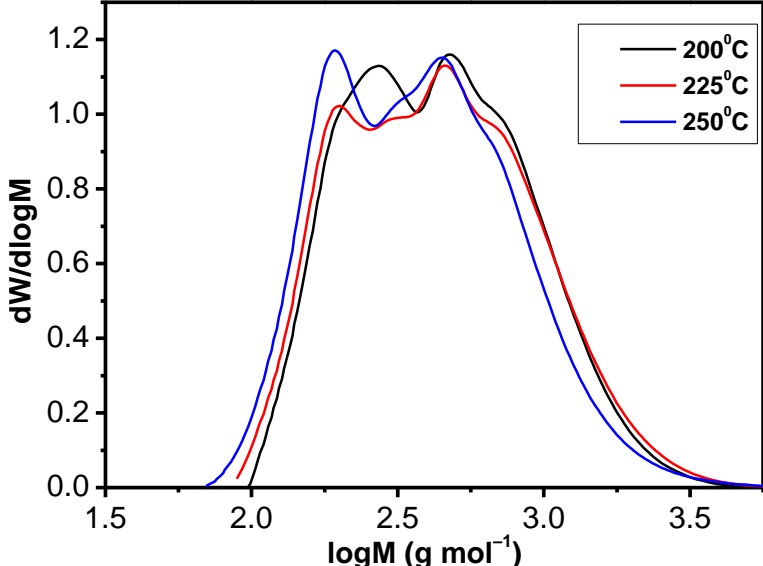

**Figure 9.** The effect of temperature of spruce wood RCF for 3 h over the 3RS450 catalyst on the molecular weight distribution of the liquid products.

**Table 6.** Effect of temperature of spruce wood RCF over the 3RS450 catalyst for 3 h on the molecular weight characteristics of the liquid products.

| Temperature, °C | $M_n$ [a], g mol$^{-1}$ | $M_w$ [b], g mol$^{-1}$ | PD [c] |
|---|---|---|---|
| 200 | 350 | 565 | 1.614 |
| 225 | 337 | 578 | 1.715 |
| 250 | 299 | 502 | 1.679 |

[a] The number average molecular weight; [b] the weight average molecular weight; [c] the polydispersity $M_w/M_n$.

*2.4. The Yield and Composition of the Products of Spruce Wood RCF over the 3RS450 Catalyst Depending on Time*

We studied the effect of the time of RCF of spruce wood at 225 °C over the 3RS450 catalyst on the product yield. As the process time lengthens from 3 to 4.5–6 h, the yields of the liquid and solid products decrease from 36 to 29 wt% and from 49.5 to 46 wt%, respectively, and the yield of the gaseous products increases from 10.6 up to 14.3 wt%; whereas the lignin and hemicelluloses contents in the solid product decrease from 8.6 to 6.8 wt% and from 7.0 to 6.0 wt%, respectively, while the cellulose content increases from 84.4 to 87.2 wt% (Table 7).

**Table 7.** The effect of time of spruce wood RCF at 225 °C over the 3RS450 catalyst on the products yield and composition.

| Reaction Time, h | Product Yield, wt% | | | | | Solid Product Composition, wt% | | |
|---|---|---|---|---|---|---|---|---|
| | Solid | Liquid | Gaseous | Phenolic Monomers [a] | Polyols [b] | Hemicelluloses | Lignin | Cellulose |
| 3.0 | 49.5 | 36.0 | 10.6 | 26.0 | 18.6 | 7.0 | 8.6 | 84.4 |
| 4.5 | 48.2 | 31.0 | 11.7 | 22.6 | 15.5 | 6.2 | 7.6 | 86.2 |
| 6.0 | 46.0 | 28.8 | 14.3 | 21.4 | 13.6 | 6.0 | 6.8 | 87.2 |

[a] The yield from the lignin weight in wood; [b] the yield from the polysaccharide weight in wood.

The yields of monomer methoxyphenols in the liquid products decreases progressively from 26 to 21.4 wt% with lengthening of the spruce wood RCF time from 3 to 6 h. It seems like the yield of these compounds decreases due to the contribution of condensation reactions of the monomer compounds. The increasing process time affects also the composition of phenolic monomers: the yields of 4-propenylguaiacol (3) and 4-propylguaiacol (2) decrease and 4-propanolguaiacol (4) increase (Figure 10A).

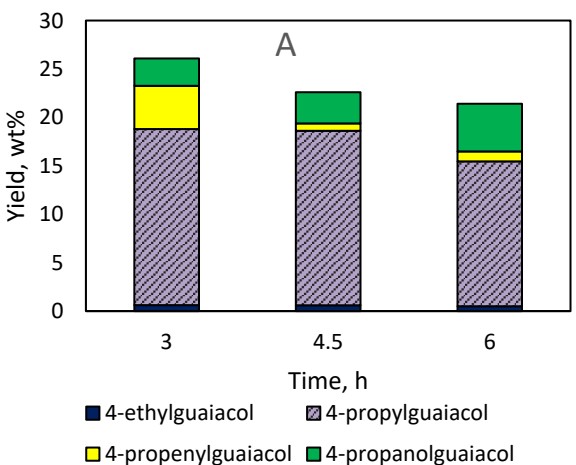
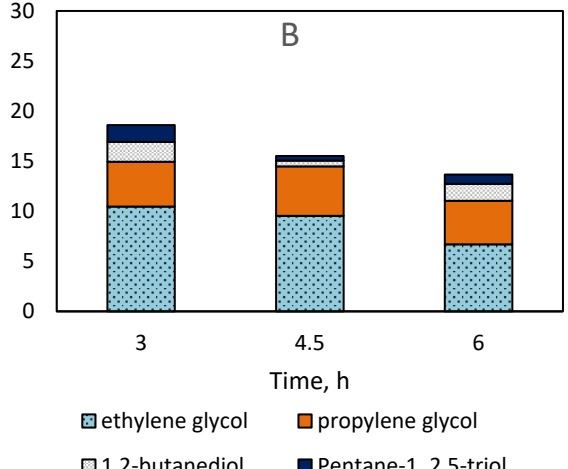

**Figure 10.** The effect of time of spruce wood RCF on the yield of phenolic monomers (**A**) and polyols (**B**) over the 3RS450 catalyst at 225 °C.

The yield of glycols, mainly ethylene glycol and propylene glycol, in the liquid products decreases more sharply than the yields of phenolic monomers (from 18.6 to 13.7 wt%) with the increasing RCF time (Figure 10B). An increase in the yield of the gaseous products with the spruce wood RCF time can be accounted for by the contribution of the hemicelluloses hydrocracking reaction to produce CO [44] (Figure 11).

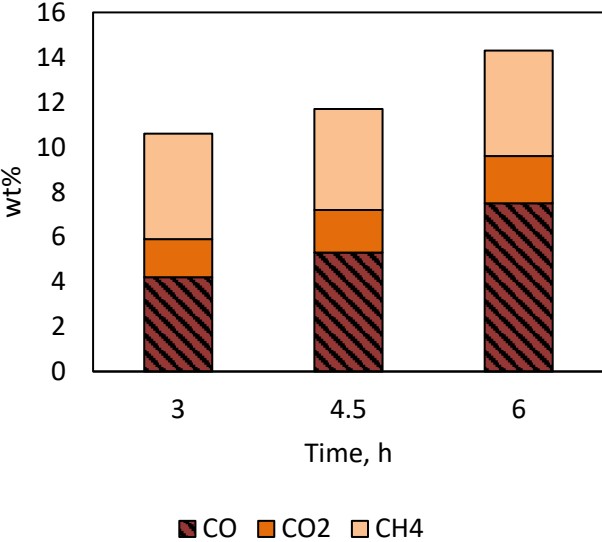

**Figure 11.** The effect of time of spruce wood RCF over the Ru/C catalyst at 225 °C on the yield and composition of the gaseous products.

The GPC data reveal a narrowed high-molecular shoulder on the MWD curve of the liquid products and decreasing polydispersity PD (from 1.715 to 1.623); this occurs due to a decrease in the average molecular weight $M_w$ of the products from 578 to 526 g/mol at a longer RCF time. The intensity of the peaks corresponding to the mono- and dimeric products of the lignin transformation also reduces to indicate a decrease in their yields (Table 8, Figure 12).

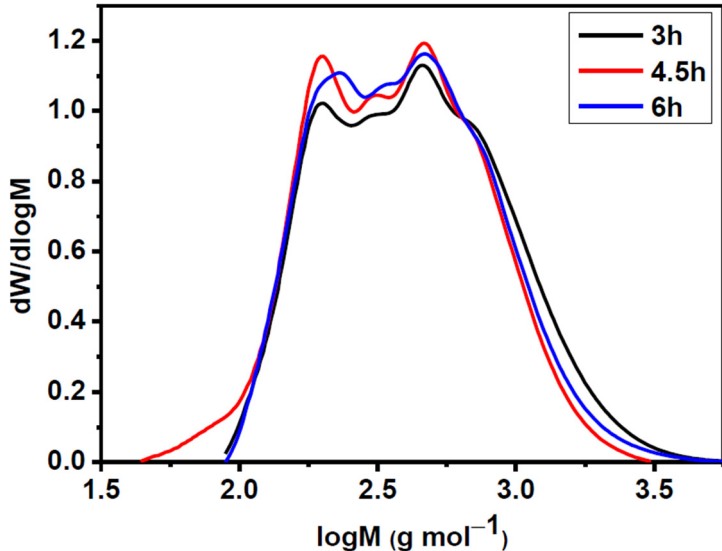

**Figure 12.** The effect of time of spruce wood RCF over the Ru/C catalyst at 225 °C for 3 h on the molecular weight distribution of the liquid products.

**Table 8.** The effect of time of spruce wood RSF over the 3RS450 catalyst at 225 °C for 3 h on the molecular weight characteristics of the liquid products.

| Process Time, h | $M_n$ [a], g mol$^{-1}$ | $M_w$ [b], g mol$^{-1}$ | PD [c] |
|---|---|---|---|
| 3.0 | 337 | 578 | 1.715 |
| 4.5 | 329 | 545 | 1.626 |
| 6.0 | 324 | 526 | 1.623 |

[a] The number average molecular weight; [b] the weight average molecular weight; [c] the polydispersity $M_w/M_n$.

## 2.5. Effect of Ru/C Catalyst Acidity on the Yield and Composition of the Spruce Wood RCF Products

Acidic conditions are known to promote hydrolysis of ether bonds. The most prominent event in acid-catalyzed lignin chemistry is the cleavage of b-O-4 ether bonds. Nevertheless, acidic media affect the lignin structure by facilitating both depolymerisation (i.e., acidolysis) and repolymerisation [4]. In order to check these impacts, we used bifunctional powder catalysts based on acidic supports.

The acid-base properties of the ruthenium catalysts were changed by oxidizing the carbon support Sibunit at 400 and 450 °C. The acidity of the catalysts determined from the $pH_{pzc}$ (Table 9) vary from 6.89 to 8.01. The highest yields of the liquid products (36.0 and 32.5 wt%, respectively) were obtained over the 3RS450 and 3RS400 bifunctional catalysts at $pH_{pzc}$ values of 7.12 and 6.89. The yield of the solid product decreases progressively from 57.4 to 49.5 wt% with decreasing $pH_{pzc}$. The solid products contain less lignin and hemicelluloses and more cellulose, in contrast to the case of the monofunctional ruthenium catalyst with the weak-alkaline properties (pH 8.01) (Table 9).

**Table 9.** The effect of $pH_{pzc}$ value of the Ru catalysts on the yield and composition of products of spruce wood RCF at 225 °C for 3 h.

| Catalyst | $pH_{pzc}$ | Product Yield, wt% | | | | Solid Product Composition, wt% | | | |
|---|---|---|---|---|---|---|---|---|---|
| | | Solid | Liquid | Gaseous | Phenolic Monomers [b] | Polyols [c] | Hemicelluloses | Lignin | Cellulose |
| 3RS450 | 6.89 | 49.5 | 36.0 | 10.6 | 26.0 | 18.6 | 7.0 | 8.6 | 84.4 |
| 3RS400 | 7.12 | 54.3 | 32.5 | 9.0 | 30.0 | 13.9 | 14.8 | 7.6 | 77.6 |
| 3RS [a] | 8.01 | 57.4 | 30.0 | 9.0 | 22.0 | 12.5 | 14.1 | 9.8 | 76.1 |

[a] The original nonoxidized support; [b] the yield from the lignin weight in wood; [c] the yield from the polysaccharide weight in wood.

The yields of phenolic monomers and polyols (Figure 13A,B) are essentially affected by the catalyst acidity. On oxidation of the carbon support at 400 °C followed by $pH_{pzc}$ decrease to 7.12, the yield of the monomer phenolic compounds increases from 22 to 30 wt%. Oxidation at higher temperature (450 °C) results in a decrease in $pH_{pzc}$ to 6.89 and a decrease in the yield of these compounds to 26 wt%. The yield of polyols increases with a decrease in the catalyst $pH_{pzc}$; it reaches 18.6 wt% over the most acidic catalyst 3RS450 with $pH_{pzc}$ 6.89. The increase in the yield of polyols seems to cause an increase in the efficiency of the glycosidic bond hydrolysis in hemicelluloses and/or amorphous cellulose with increasing acidity of the catalyst. [45]. (Table 9).

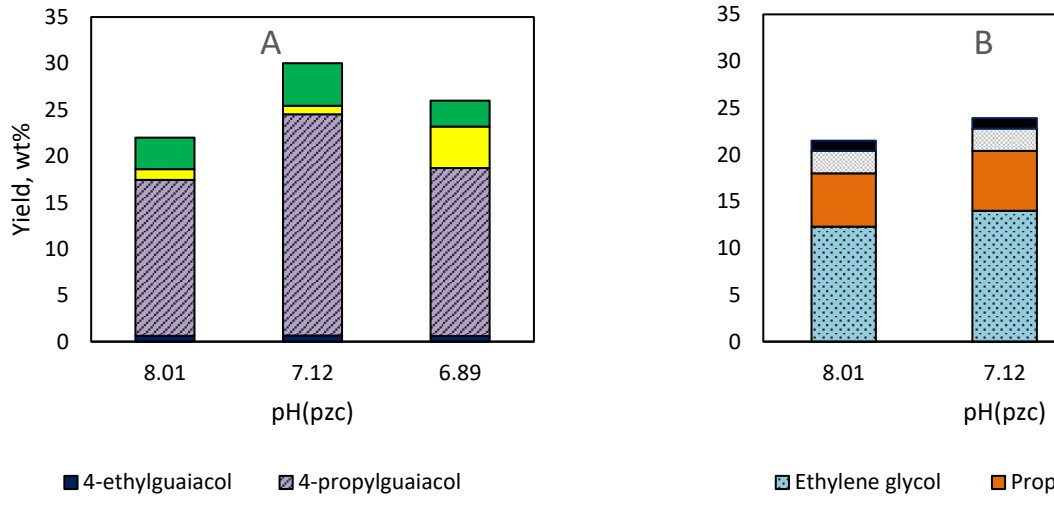

**Figure 13.** The effect of pH$_{(pzc)}$ of the catalysts on the yields of phenolic monomers (**A**) and polyols (**B**) at spruce wood RCF at 225 °C for 3 h.

### 2.6. Characterization of the Solids Obtained from RCF of Spruce Wood

The structure of the solid products of spruce wood hydrogenation was studied using infrared (IR) spectroscopy and X-ray diffraction (XRD) techniques. IR spectra of the initial wood and the solid products of its hydrogenation are shown in Figure 14. The IR spectrum of the initial spruce wood is a sum of absorption bands of its main structural components and includes the bands characteristic of bonds between cellulose, lignin, and hemicellulose macromolecules (Figure 14) [46]. The absorption band at 1737 cm$^{-1}$ is assigned to stretching vibrations of the C=O group in the uronic acid ester monomer of hemicelluloses [47]. In the solid products of the RCF, the intensity at 1737 cm$^{-1}$ almost disappears. The absorption bands at 1606 and 1510 cm$^{-1}$ in the IR spectra of spruce wood and the solid products of its RCF correspond to skeletal vibrations of syringyl and guaiacyl aromatic rings [48]. These less intense bands, as well as band at 1269 cm$^{-1}$ (C-O stretching of guaiacyl unit) [48], in the solid RCF products argue for a decrease in the lignin content.

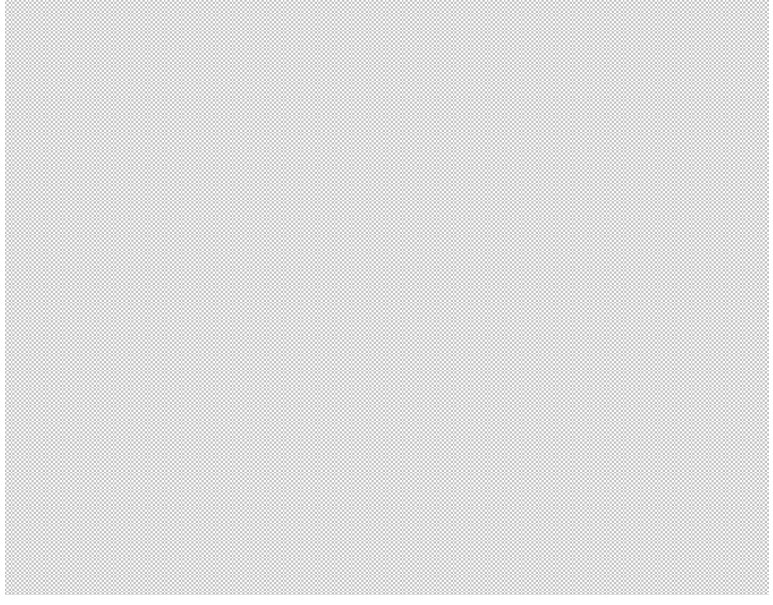

**Figure 14.** IR spectra for the initial spruce wood samples (1) and the solid products of wood RCF over the 3RS450 (2) and 3RS400 (3) catalysts.

The absorption bands at 1500–900 cm$^{-1}$ are assigned to vibrations of C–H bonds in the methyl and methylene groups, C–O and O–H bonds, the glycosidic bond, and the cellulose glucopyranose ring [49].

Figure 15 shows XRD patterns of the samples of the initial spruce wood and solid products of its hydrogenation.

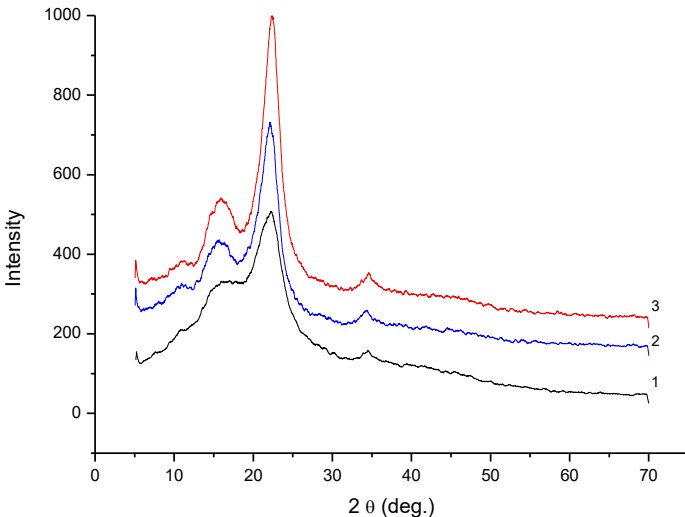

**Figure 15.** X-ray diffraction patterns for the initial spruce wood sample (1) and solid products of RCF over the 3RS400 (2) and 3RS450 (3) catalysts.

XRD patterns of all the samples under study contain two intense peaks with maxima at 2θ angles of 22.2° and 15.6° (Figure 15). They correspond to reflections of atoms at the (002) plane and superimposed reflections of atoms at the (101) and (10$\bar{1}$) planes of the cellulose crystal lattice [50]. A pronounced peak at a 2θ diffraction angle of ~22.2° is a criterion for the cellulose crystallinity and characterizes the fraction of densely packed cellulose molecules [51].

The crystallinity index of the initial spruce wood is 0.47. The solid products of RCF of the wood have the higher crystallinity indices as compared with those of the initial spruce wood. This relates to the removal of a significant part of amorphous carbohydrates during the wood hydrogenation (Table 10).

**Table 10.** Crystallinity indices for the spruce wood samples and solid products of wood RCF.

| Sample | Crystallinity Index of the Solid Product |
|---|---|
| Spruce wood | 0.47 |
| Catalyzed solid hydrogenation products (RS400) | 0.71 |
| Catalyzed solid hydrogenation products (RS450) | 0.78 |

The highest crystallinity index (0.78) is characteristic of the solid product of the catalytic hydrogenation of spruce wood over the more acidic RS450 bifunctional catalyst. The composition and structure of the solid product corresponds to the parameters of commercial microcrystalline cellulose with a crystallinity index of 0.75 [52].

### 2.7. Optimal Conditions for the Reductive Catalytic Fractionation of Spruce Wood

To be an effective biomass processing strategy for fuels and commodity chemical production, RCF should provide high levels of lignin and hemicelluloses extraction as well as depolymerization but minimize cellulose dissolution and decomposition. A trade-off between these parameters exists based on the data shown in the present study. A scenario aimed at achieving an optimal balance between the removal of lignin and the preservation

of all carbohydrates in the pulp seems difficult to realize due to the fact that lignin and hemicellulose are intertwined in the cell wall.

The conditions for the RCF process providing the optimal trade-off between the yields of phenols, polyols, and cellulose form spruce wood are determined (Figure 16).

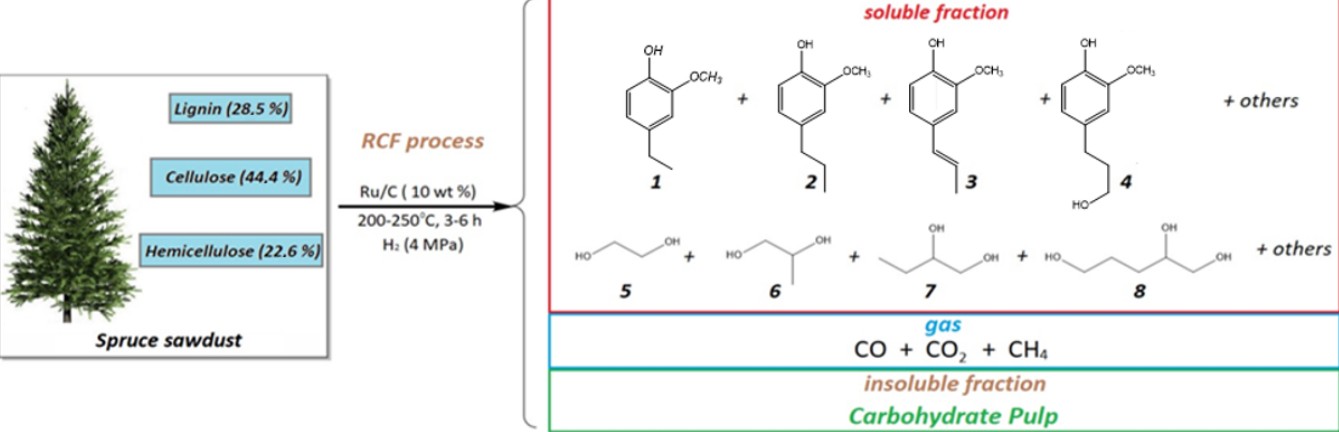

**Figure 16.** The main products of the reductive catalytic fractionation of spruce wood.

Trends in the influence of various process parameters (temperature, time, content and dispersion of ruthenium, catalyst acidity, catalyst particle size) on the yield and composition of liquid, solid, and gaseous products of spruce wood RCF have been revealed. The yield of monomeric phenols and polyols in the liquid products increases with the hydrogenation temperature elevation; an increase in the process time over 3 h reduces their yield. Thus, the highest yield of phenolic monomers and polyols are achieved at the process temperature of 250 °C and a process time of 3 h. The yields of phenolic and polyolic monomers increase with increasing content and dispersion of ruthenium in the catalyst as well as with increasing catalyst acidity.

Summarizing the results, we can conclude: the most efficient spruce wood RCF catalysts, that provide the highest yield of monomeric phenols and polyols, are the 3RS400 and 3RS450 bifunctional catalysts, which contain 3 wt% of Ru with a dispersion of 0.94, possess high acidity (pH 7.12 and 6.89, respectively), and have a particle size of 56–94 μm.

Figure 17 shows the Sankey diagram for the material balance of the spruce wood RCF process (3 g loaded into the reactor) calculated for the experiment over the 3RS450 catalyst (10 wt% of the wood weight). The initial pre-deresined spruce wood sample contained 1.407 g of cellulose, 0.869 g of lignin, and 0.719 g of hemicelluloses. The reductive catalytic fractionation in ethanol at 225 °C for 3 h yielded 1.470 g of microcrystalline cellulose (49.0% of the wood weight), 1.049 g of the liquid products (36.0% of the wood weight), and 0.318 g of gases (10.6% of the wood weight). After extraction of the liquid products with dichloroethane and water, 0.413 g of the aqueous phase containing 0.392 g of glycols (18.6% of the polysaccharide weight in the wood) and 0.636 g of the dichloromethane phase containing 0.233 g of phenolic monomers (26.0% of the lignin weight in the wood) were obtained (Figure 17). The total loss of these substances during the extraction and pyrogenetic water was 0.163 g (5.4% of the wood weight).

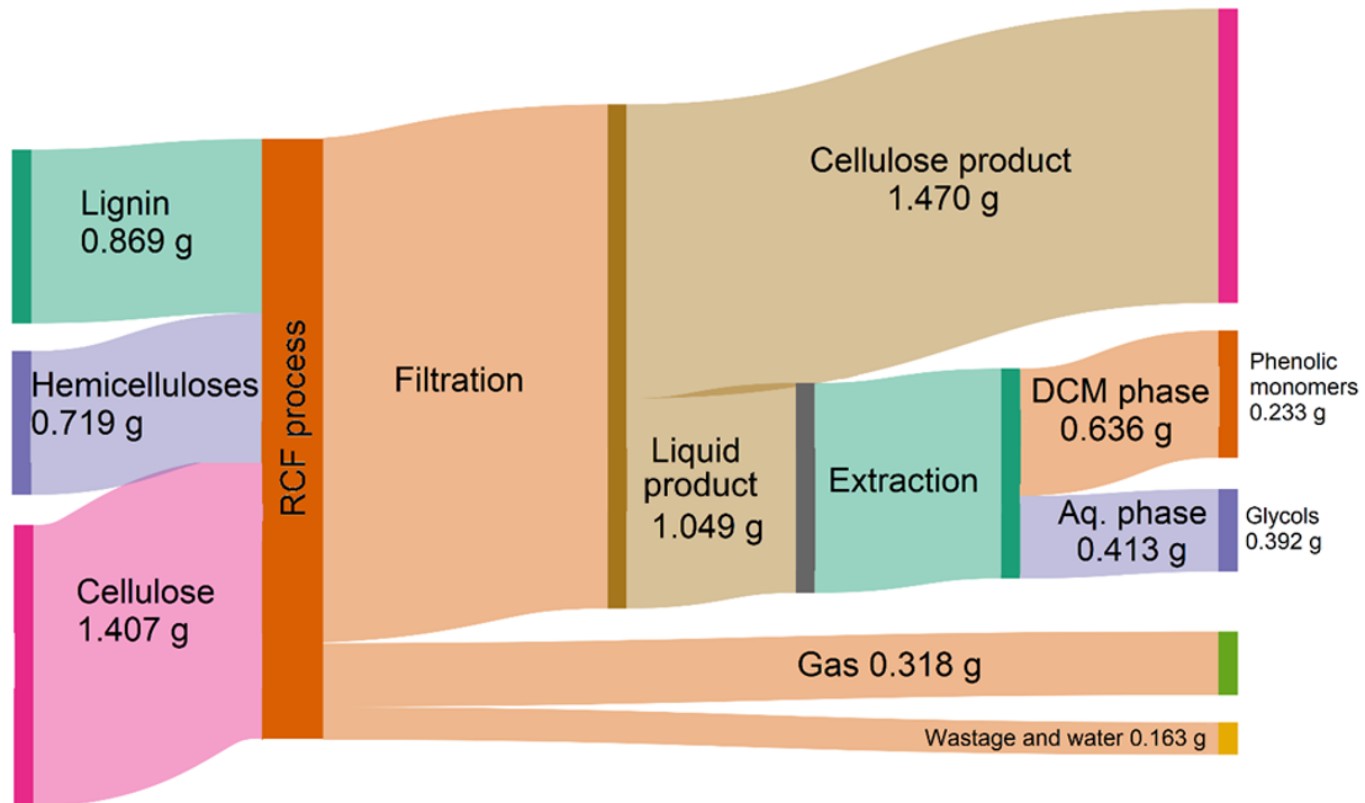

**Figure 17.** Sankey diagram for the material balance of the reductive catalytic fractionation of spruce wood over the 3RS450 catalyst.

Thus, the optimum trade-off between the yields of the liquid products enriched in phenolic monomers and polyols is obtained at the spruce wood hydrogenation temperature of 225 °C and the time of 3 h over the the highly-acidic 3% Ru/C bifunctional catalyst.

## 3. Materials and Methods

### 3.1. Materials and Reagents

Siberian spruce (*Pícea ováta*) wood (Krasnoyarsk region) used in this study contained cellulose (44.4), lignin (28.6), hemicelluloses (22.6), extractives (3.8), and ash (0.6%) from the weight of absolutely dry wood. The wood was ground into particles smaller than 1 mm in size, successively deresined with petroleum ether and acetone by a conventional ANSI/ASTM D 1105 method, and then dried at 24 °C to a constant moisture content (4.3 wt%).

In this study, we used AO RFK ethyl alcohol (95%), $Ru(NO)(NO_3)_3$ (31.3% of Ru, Alfa Aesar, Ward Hill, MA, USA), $-N$, O bistrifluoroacetamide (BSTFA + TMCS 99:1, Sigma-Aldrich, Burlington, MA, USA), and ionol (>99%, Sigma-Aldrich, Burlington, MA, USA).

### 3.2. Preparation and Study of the Ru/C Catalysts

Carbon supports containing acid sites were prepared by oxidizing a Sibunit commercial mesoporous carbon material with an oxygen–argon mixture containing 20 vol% of $O_2$ in the presence of water vapor (saturation at 90 °C, vapor pressure 70.1 kPa) at temperatures of 400, 450, and 500 °C for 2 h using the technique described elsewhere [53].

One percent Ru/C and 3% Ru/C catalysts were prepared by incipient wetness impregnation of a carbon support with the $Ru(NO)(NO3)_3$ aqueous solution followed by drying at room temperature for 2–3 h and then at 60 °C for 12 h. The active component was reduced in flowing hydrogen (30 mL/min) at 300 °C and the rate of 1 °C/min for 2 h. After cooling down to room temperature in the hydrogen atmosphere, the catalysts were passivated using 1% $O_2$ in argon as described in [54].

The texture characteristics of the samples were determined from the $N_2$ adsorption isotherms using a Micromeritics ASAP-2020 Plus analyzer (US) at 77 K. The size distribution of ruthenium particles in the ruthenium catalysts was determined using a Hitachi HT7700 transmission electron microscope (Toranomon Minato-Ku, Japan, 2014) at the accelerating voltage of 110 kV and resolution of 2 Å. The particle size distribution histograms were obtained by the statistical (500–800 particles) processing of electron microscopy images. The linear-average ($<d_l>$) and weight-average ($<d_s>$) diameters of deposited particles were calculated using the formulas [55]:

$$<d_l> = \Sigma d_i / N, \, <d_s> = \Sigma d_i^3 / \Sigma d_i^2,$$

where $d_i$ is the diameter of the deposited particle and $N$ is the total number of particles.

The ruthenium dispersion $D_{Ru}$ in the catalysts was calculated as:

$$D_{Ru} = 6 \cdot \frac{M_{Ru}}{a_{Ru} \cdot \rho \cdot N_0 \cdot \langle d_s \rangle} \tag{1}$$

where $M_{Ru}$ = 0.101 kg/mol is the ruthenium molar mass, $\rho$ = 12,410 kg/m$^3$ is the density of ruthenium metal, $a_{Ru}$ = 6.13; $10^{-20}$ m$^2$ is the average effective area of a metal atom on the surface, $N_0$ is the Avogadro number, and $d_s$ is the weight-average ruthenium particle diameter [55].

The acid-base properties of the catalysts were investigated by finding the point of zero charge (pzc) by the Sörensen–de Bruin method [41]. This method was shown [53] to be suitable for comparing the acid properties of carbon materials. The results obtained by this method correlate well with the results of XPS and acid-base titration [53], and the technique does not require expensive instruments or lengthy experiments compared to other methods. Ten mL of distilled water was poured in a potentiometric cell and the test sample was successively added as small (0.01-g) portions in specified (5–10 min) intervals under continuous stirring with a magnetic stirrer until the constant glass electrode potential was reached [41].

### 3.3. Catalytic Hydrogenation of Spruce Wood

The process of hydrogenation of crushed deresined wood was studied in a 300 mL ChemRe SYStem R-201 autoclave (Gwanyang–dong, Korea) (see Figure 18). Fifty mL of ethanol, 5.0 g of sawdust, and 0.5 g of the ruthenium catalyst were loaded into the reactor. Then, the autoclave was hermetically sealed, purged with argon to remove air, and filled with hydrogen to a pressure of 4 MPa. The hydrogenation was carried out at 200, 225, and 250 °C for 3 h under continuous stirring at the speed of 800 rpm. The operating pressure in the reactor ranged from 8.6 to 9.5 MPa.

After completion of the reaction and cooling of the reaction mixture to room temperature, gaseous products were collected in a gasometer, their volume was measured, and the composition was determined by gas chromatography (GC). The mixture of the liquid and solid products was quantitatively discharged from the autoclave by washing with ethanol and separated by filtration; then, the solid product was washed with ethanol until the solvent discoloration and dried to a constant weight at 80 °C. Ethanol was removed from the liquid product by distillation on a rotary evaporator and the liquid product was brought to a constant weight by drying under vacuum (1 mmHg) at room temperature. The yields of the liquid ($a_1$), solid ($a_2$), and gaseous ($a_3$) products and the spruce wood conversion ($Y_1$) were calculated using the equations:

$$a_1 = \frac{m_l(\text{g})}{m_{init}(\text{g})} \times 100\% \tag{2}$$

$$a_2 = \frac{m_s(\text{g}) - m_{cat}(\text{g})}{m_{init}(\text{g})} \times 100\% \tag{3}$$

$$a_3 = \frac{m_g(\text{g})}{m_{init}(\text{g})} \times 100\% \tag{4}$$

$$Y_1 = \frac{m_{init}(\text{g}) - m_s(\text{g})}{m_{init}(\text{g})} \tag{5}$$

where $m_l$ (g) is the liquid product weight, $m_{init}$ (g) the weight of the initial wood, $m_s$ (g) the solid product weight, $m_{cat}$ (g) is the catalyst weight, and $m_g$ (g) the gaseous products weight.

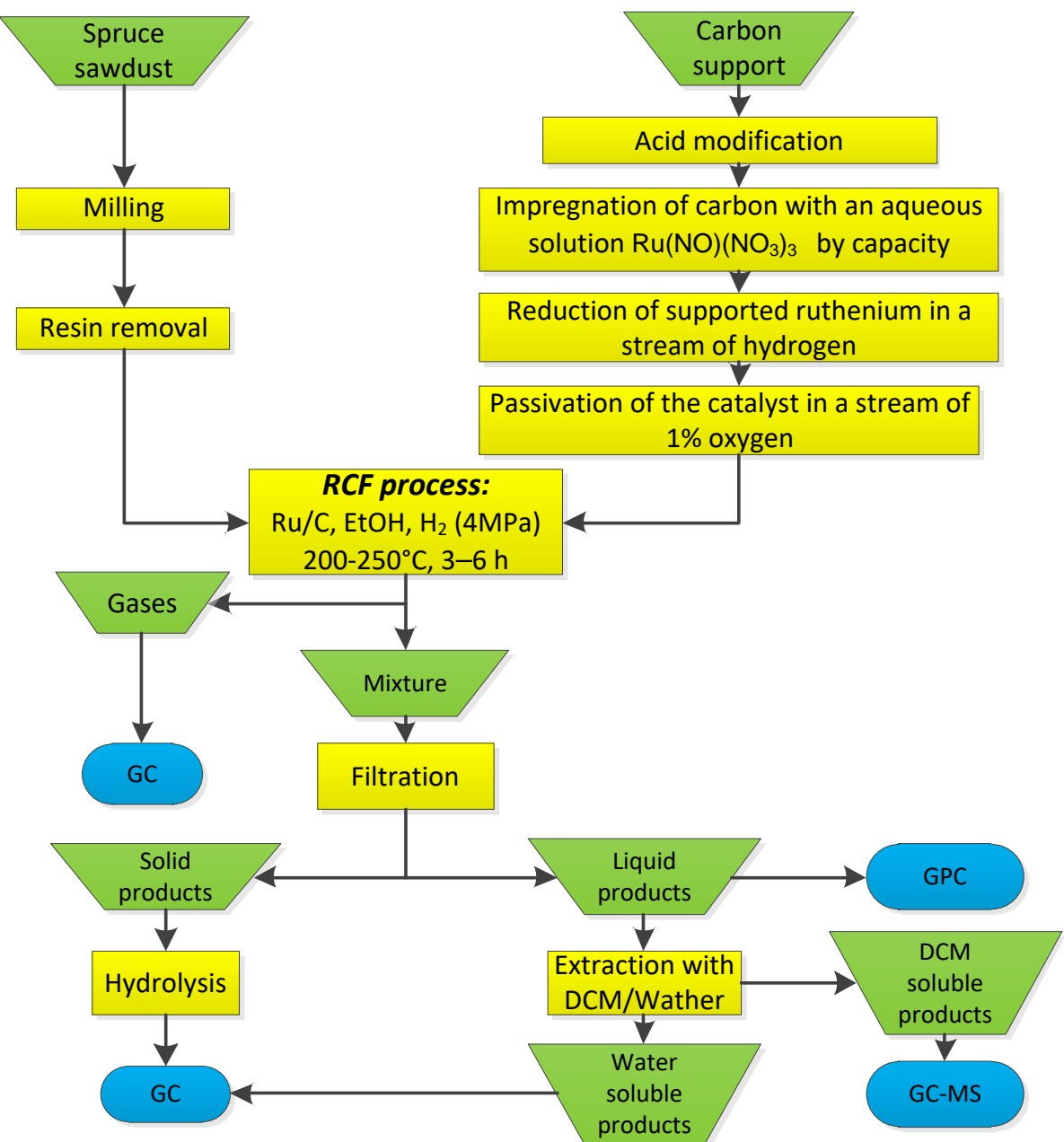

**Figure 18.** Overview of process flow diagram of catalytic hydrogenation of spruce wood.

### 3.4. Study of the Spuce Wood RCF Products

The ethanol-soluble liquid products of wood hydrogenation were extracted with a mixture of dichloromethane and water to separate into the aqueous and organic fractions (see Figure 17). The aqueous fraction was silylated by evaporating 1 mL of the solution to dryness and adding 100 µL of pyridine and 100 µL of −N,O-bis(trimethylsilyl)trifluoroacetamide (BSTFA); then, 10 µg of the internal standard (ionol) was introduced and kept at 70 °C for

1 h. After that, the organic and aqueous phases were analyzed by gas chromatography–mass spectrometry (GC–MS) using an Agilent 7890A chromatograph with an Agilent 7000A Triple Quad selective mass detector and an HP-5MS capillary column (30 m) at temperatures programmed within 40–250 °C. The compounds were identified using the NIST MS Search 2.0 database. Propyl guaiacol, propenyl guaiacol, ethyl guaiacol (Sigma-Aldrich) were used as standard substances for quantifying the yield of monomers from lignin. The response factor for each standard compound was defined relative to the internal standard [56]. Phenanthrene was used as an internal standard. Ethylene glycol, propylene glycol, 1,2-butanediol and glycerol (Acros Organics, Fair Lawn, NJ, USA) were used as standards for the quantitative determination of products in the aqueous phase.

The molecular weight distribution of the liquid wood RCF products, weight average molecular weight $M_w$, number average molecular weight $M_n$, and polydispersity PD of the liquid product samples were determined by gel permeation chromatography (GPC) using an Agilent 1260 Infinity II Multi-Detector GPC/SEC System with the triple detection using a refractometer, a viscometer, and light scattering. The mixtures were separated in a PLgel Mixed-E column using tetrahydrofuran stabilized with 250 ppm BHT as a mobile phase. The column was calibrated using Agilent polydisperse polystyrene standards (Santa Clara, USA). The eluent feed rate was 1 mL/min and the volume of the injected sample was 100 μL. Before the analysis, the samples were dissolved in THF (1 mg/mL) and filtered through a 0.45-μm Millipore PTFE membrane filter. The data obtained were collected and processed using the Agilent GPC/SEC MDS software ver. 2.2.

The composition of the gaseous products was determined by GC using a Chromatec Crystal 2000 M chromatograph (Russia) with a thermal conductivity detector in a helium flow at the flow rate of 15 mL/min and a detector temperature of 170 °C. The CO and $CH_4$ analysis was carried out in a NaX zeolite column (3 m × 2 mm) in the isothermal mode at 60 °C. The analysis of $CO_2$ and hydrocarbon gases was carried out in a Porapak Q column in the following mode: 60 °C for 1 min and then elevation of temperature to 180 °C at the rate of 10 °C/min.

The solid wood product was analyzed for hemicelluloses, cellulose, and lignin contents. The residual lignin content was determined by hydrolysis with 72% $H_2SO_4$ at 98 °C [57]. The composition and concentration of monosaccharides in the solution obtained by hemicelluloses hydrolysis with 4% sulfuric acid were determined by GC [57]. The cellulose content was calculated from the difference between the wood weight (or the solid residue) and the hemicelluloses and lignin contents.

The GC study was carried out with a Varian-450 GC gas chromatograph (Varian, Inc., Palo Alto, CA, USA), a flame ionization detector, and a VF-624ms capillary column with a length of 30 m, an inner diameter of 0.32 mm, and helium as a gas carrier at an injector temperature of 250 °C. Before the analysis, the solution was derivatized to produce trimethylsilyl derivatives according to the procedure described elsewhere [58]. Sorbitol was used as an internal standard. The peaks were identified from retention times of the tautomeric forms of monosaccharides.

The elemental compositions of wood and the products of its conversion were determined using a CHNSO VARIO EL CUBE analyzer (Elementar, Germany).

X-ray diffraction (XRD) analysis was carried out using a PANalyticalX'Pert Pro (PANalytical, EA Almelo, The Netherlands) spectrometer with CuKα radiation (λ = 0.54 nm). The analysis was performed in the angle 4 range of 2θ = 5–70° in 0.1° intervals on the powder sample in a 2.5-cm diameter cuvette. The crystallinity index was calculated as the ratio between the crystalline peak intensities $I_{002}$–$I_{AM}$ and the total intensity $I_{002}$ after subtraction of the background signal:

$$CI = \frac{I_{002} - I_{AM}}{I_{002}}, \qquad (6)$$

where $I_{002}$ is the height of peak 002 and $I_{AM}$ is the height of the minimum between peaks 002 and 101 [51].

## 4. Conclusions

Reductive catalytic fractionation (RCF) of lignocellulose offers a biorefinery technology which integrates both lignin and carbohydrates valorization. During this process, lignin is solvolytically extracted and simultaneously depolymerized via hydrogenolysis. In this study, we propose the RCF of spruce wood over a bifunctional catalyst Ru/C in the medium of ethanol and molecular hydrogen to produce monomer phenolic compounds from lignin, polyols from hemicelluloses, and microcrystalline cellulose.

TIe influence of the Ru/C bifunctional catalyst characteristics was assessed. The ruthenium catalysts were characterized by transmission electron microscopy. The solid, liquid, and gaseous products of the reductive fractionation of spruce wood biomass were characterized by IR spectroscopy, XRD analysis, GC-MS, GPC, GC, $N_2$ adsorption methods and by elemental and chemical analysis. The results show the particular effect of the carbon support acidity, catalyst grain size, content and dispersion of Ru on the effectiveness of lignin and hemicelluloses extraction and the yields of liquid and gaseous products. The most efficient catalysts for RCF of spruce wood providing high yields of monomer phenols, glycols, and solid product with content of cellulose up to 90 wt% bear 3 wt% of Ru with a dispersion of 0.94 based on an acidic oxidized graphite-like carbon support Sibunit®, and having a grain size of 56–94 μm.

**Author Contributions:** Conceptualization, O.P.T. and B.N.K.; methodology, A.V.M., A.S.K., A.M.S. and S.V.B.; formal analysis A.M.S., V.V.S., A.V.M., Y.N.M. and S.V.B.; investigation, A.V.M., A.S.K., V.V.S. and A.M.S.; data curation A.S.K., A.V.M. and Y.N.M.; writing—original draft preparation, A.V.M. and S.V.B.; writing—review and editing, O.P.T. and B.N.K.; visualization, A.V.M. and V.V.S.; supervision, O.P.T. and B.N.K.; project administration, O.P.T.; funding acquisition, O.P.T. All authors have read and agreed to the published version of the manuscript.

**Funding:** This study was supported by the Russian Science Foundation, project no. 21-73-20269, https://rscf.ru/project/21-73-20269/ (accessed on 1 November 2022).

**Data Availability Statement:** Data available on request from the authors.

**Acknowledgments:** This study was conducted using the equipment of the Krasnoyarsk Regional Centre for Collective Use, Krasnoyarsk Scientific Center, Siberian Branch of the Russian Academy of Sciences.

**Conflicts of Interest:** The authors declare that they have no conflict of interest.

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
