# Peer review of "Reductive Catalytic Fractionation of Spruce Wood over Ru/C Bifunctional Catalyst in the Medium of Ethanol and Molecular Hydrogen"

_catalysts, doi:10.3390/catal12111384_

Round 1

Reviewer 1 Report

The work presented by Taran et al. is of great interest and it is well presented and articulated. However, some points should be clarified before to be accepted:

1) Lines 47-52: Please retype these sentences, they are not very clear.

2) Lines 53-55: It is not clear the connection between the organosolv delignification, present in the above paragraph, and the oxidative/reductive catalytic fractionation of biomass. Clarify.

3) Line 57: Which are the products obtained from oxidative catalytic fractionation?

4) Line 59: "reducing agent", Hydrogen or others?

5) Line 61: Which is the fate of hemicellulose in this process?

6) Line 98: "except for the sample oxidized at 450 °C", why?

7) Did the authors have the standards for the quantification of the products present in DCM and water?

8) Line 133: Maybe the authors mean solid product. Why “solid carbohydrate product” if there is also lignin?Also in Tables they reported "solid product composition".

9) Lines 160-161: Why? Please justify this observation.

10) The authors decided to focus their attention on the products reported in Figure 2 or they are the only products obtained? From GC-MS analysis other products have been observed at least as traces?

11) In order to better observe the data reported in Table 3 may be a van Kravelen diagram should be introduced.

12) Lines 181-183: Why? Ruthenium have catalysed the hydrogenation of carbon oxides? Justify.

13) Figure 5: The curve of 3% Ru is not evident, please change the type of line.

14) Table 5: The notes are missing.

15) Lines 300-302: Cannot be simply promoted the reduction of CO to methane?

16) Figure 11:The curve of 6h is not evident, please change the type of line.

17) Table 9: The notes are missing.

18) Line 344: Is not 450 °C?

19) Lines 344-345: Why this trend?

20) I think that Paragraph 3.6 should be the last and include also the Sankey diagram and relative comments.

21) Which is the assignment of the band at 1269 cm-1?

22) Lines 413-414: Cellulose or hemicellulose? Hemicellulose is amorphous and its removal can contribute to increasing the crystallinity of the solid.

23) Which instrument has been adopted for the determination of the crystallinity index?

24) The references in the Comment paragraph should be avoided. The included references could be moved in the introduction in order to highlight the relevance of the compounds produced in the paper.

Author Response

Dear Reviewer, thank you for your helpful comments, which allowed us to improve the manuscript. 

Please, see attached file.

Reviewer 2 Report

Dear Editors and authors,

In this manuscript, the authors present an interesting work dealing with reductive catalytic fractionation, aiming at the production of value-added products in a biorefinery technology. For addressing such a topic, they use a Ru/C catalytic system and different characterization methods, such as Nitrogen adsorption isotherms, TEM, PZC, GC, and elemental analysis composition, and the resulting work has interesting information. The topic is quite important, and I recommend publication after minor revision, following my comments below.

1-      Please, change all subtopics numbers on topic 2 (Results)

2-      Lines 98-101- The authors could explain the decrease in the surface area according to the suggested references (doi.org/10.3390/catal8080340 ; doi.org/10.1590/1980-5373-MR-2022-0143 )

3-  Table 3- Does the oxygen content was calculated by difference? Why the total composition of the original spruce wood is different from 100%?

4-      Please change “asid” for “acid” in figure 17.

Author Response

Dear Reviewer, thank you for your helpful comments, which allowed us to improve the manuscript.

1) Please, change all subtopics numbers on topic 2 (Results)

Answer: Thank you. Corrected.

2- Lines 98-101- The authors could explain the decrease in the surface area according to the suggested references (doi.org/10.3390/catal8080340 ; doi.org/10.1590/1980-5373-MR-2022-0143 )

Answer: Thank you! We can see the explanation in the first: «As expected, the surface area and the total pore volume decreased with the metal loading, probably as a result of partial pore blockage by the metal» and in the second article «Metal loading onto a carbon support generally results in a decrease in the surface area and total pore volume of the matrix, suggesting a partial pore blockage by the metal particles». Both mirrors our explanation provided in our paper which is «The deposition of ruthenium onto the carbon support surface also leads to a decrease in the specific surface area and pore volume (Table 1) due to blocking of some pores of the supports by particles of the active component.». We added suggested references to support our explanation.

3) Table 3- Does the oxygen content was calculated by difference? Why the total composition of the original spruce wood is different from 100%?

Answer: Yes, it was calculated by difference. The error occurred when numbers were rounded. Corrected.

4) Please change “asid” for “acid” in figure 17.

Answer: Thank you. Corrected.

Reviewer 3 Report

The manuscript entitled “Reductive Catalytic Fractionation of Spruce Wood over Ru/C

Bifunctional Catalyst in the Medium of Ethanol and Molecular Hydrogen” provide an interesting investigation on the production of monomeric phenolic compounds from lignin. Therefore, I’m pleased to recommend its publication after suitable revision.

1. In abstract, the sentence “the role the Ru/C bifunctional catalysts characteristics”, “the role the Ru/C bifunctional catalysts characteristics” should be revised.

2. The acid‒base properties of the catalysts were investigated by finding the point of zero charge (pzc) by the Sörensen‒de Bruin method. Can the acid‒base properties of the catalysts be confirmed by other method?

3. How is the stability and reusability of catalysts? Will Ru be leached from the catalysts during the reaction? Will the acidic sites be stable?

4.  The grammar and terminology should be checked. There’re a lot of grammatical mistake and unclear expressions.

5. Related literatures should be consulted (DOI: 10.1039/D0GC02770G).

Author Response

Dear Reviewer, thanks for the very useful remarks permitted to improve the manuscript.

1) In abstract, the sentence “the role the Ru/C bifunctional catalysts characteristics”, “the role the Ru/C bifunctional catalysts characteristics” should be revised.

Answer: Thank you, corrected to: “the role of the Ru/C bifunctional catalysts characteristics.”

2) The acid‒base properties of the catalysts were investigated by finding the point of zero charge (pzc) by the Sörensen‒de Bruin method. Can the acid‒base properties of the catalysts be confirmed by other method?

Answer: It was confirmed by conventional acid-base titration technique and XPS, which we implemented in previous work [1]. The strong correlation between the point of zero charge and surface acid groups concentration was revealed. Since then, we have found the acid-base titration and XPS techniques redundant, due to sufficiency of the pHpzc to determine the difference between catalysts acidity in series.

This information was added.

  1. Taran, O. P.; Polyanskaya, E. M.; Ogorodnikova, O. L.;  Descorme, C.;  Besson, M.; Parmon, V. N., Sibunit-based catalytic materials for the deep oxidation of organic ecotoxicants in aqueous solution: I. Surface properties of the oxidized sibunit samples. Catalysis in Industry 2010, 2 (4), 381-386.

3) How is the stability and reusability of catalysts? Will Ru be leached from the catalysts during the reaction? Will the acidic sites be stable?

Answer: Ru catalysts was shown to be very stable even in hot water at >100 °C (leaching in first cycle 0.8 wt.% of Ru [2]. The goal of our paper was to elucidate the role of the Ru/C bifunctional catalysts characteristics. We did not pursue the study of catalyst reusability. It will be next step of our work.

  1. Taran, O. P.; Descorme, C.; Polyanskaya, E. M.;  Ayusheev, A. B.;  Besson, M.; Parmon, V. N., Sibunit-based catalytic materials for the deep oxidation of organic ecotoxicants in aqueous solutions. III: Wet air oxidation of phenol over oxidized carbon and Rr/C catalysts. Catalysis in Industry 2013, 5 (2), 164-174. doi: 10.1134/s2070050413020104.

4) The grammar and terminology should be checked. There’re a lot of grammatical mistake and unclear expressions.

Answer: Grammatical mistake are corrected.

5) Related literatures should be consulted (DOI: 10.1039/D0GC02770G).

Answer: Thank you! This reference was taken into account in the introduction.

Round 2

Reviewer 1 Report

The authors have improved the manuscript according to the reviewer's comments thus it is ready for publication.